# Lychee Fermented by Mixed Probiotic Strains Alleviates D-Galactose-Induced Skeletal Muscle and Intestinal Aging in Mice

**DOI:** 10.3390/foods14213684

**Published:** 2025-10-29

**Authors:** Huixian Han, Jin Tao, Xiaoyue Bai, Yizhi Jing, Zhengyuan Zhai, Junjie Luo, Wanxiang Zhang, Dan Gan, Yanling Hao

**Affiliations:** 1Key Laboratory of Precision Nutrition and Food Quality, Department of Nutrition and Health, China Agricultural University, Beijing 100193, China; hanhx0807@cau.edu.cn (H.H.); tj20011113@163.com (J.T.); j15502414985@cau.edu.cn (Y.J.); luojj@cau.edu.cn (J.L.); 2College of Food Science and Nutritional Engineering, China Agricultural University, Beijing 100083, China; b20233060558@cau.edu.cn (X.B.); zhaizy@cau.edu.cn (Z.Z.); 3Sirio Pharma Co., Ltd., Shantou 515000, China; wanxiang.zhang@siriopharma.com; 4Heyuan Branch Center of Guangdong Laboratory for Lingnan Modern Agricultural Science and Technology, Heyuan 517000, China

**Keywords:** lychee fermentation, mixed probiotics, skeletal muscle aging, intestinal aging, gut microbiota, phenolic acid, flavonoid

## Abstract

Aging-associated skeletal muscle and intestinal dysfunction is largely driven by chronic inflammation, oxidative stress, and microbiota imbalance. This study investigated the protective effects of a lychee fermentate (LF) in a D-galactose-induced aging mouse model. LF was prepared using a mixed microbial fermentation approach with *Lactiplantibacillus plantarum*, *Lacticaseibacillus casei*, *Saccharomyces cerevisiae*, and *Acetobacter pasteurianus SP021*. LF administration significantly improved muscle strength and endurance and restored muscle fiber morphology. Meanwhile, LF alleviated colonic transit impairment and downregulated senescence markers p16 and p21. H&E and AB-PAS staining showed that intervention with LF ameliorated the colonic tissue damage, preserved goblet cell populations and promoted MUC2-mediated mucus secretion, which was further confirmed by the upregulation of intestinal barrier-related proteins MUC2, ZO-1, Claudin-1, and Occludin through immunofluorescence analysis. In addition, LF reduced colonic inflammation by suppressing IL-1β, IL-6, TNF-α, CXCL1, and MCP1 expression, and mitigated oxidative stress by lowering malondialdehyde levels to 24.65 ± 3.84 nmol/mL while enhancing glutathione peroxidase and superoxide dismutase activities. Moreover, the LF restored intestinal health by modulating microbiota homeostasis, such as adjusting the Firmicutes/Bacteroidetes ratio and increasing the abundance of beneficial bacteria like Clostridia_UCG-014 and Alistipes. Metabolomic profiling indicated that the enhanced bioactivity of the LF was primarily attributed to the enrichment of phenolic acids, flavonoids and their derivatives postfermentation, including ethyl caffeate, gallic acid, kaempferol and isorhamnetin. In summary, these findings provided new insights into the potential application of LF as a functional food for mitigating skeletal muscle and intestinal aging.

## 1. Introduction

As global life expectancy increases, the aging population is expanding rapidly. According to the United Nations Population Division, the proportion of people aged 60 and over is projected to rise by 10.2% between 2010 and 2050, while those aged 80 and above will increase by 2.5% [1]. The physiological impact of aging is particularly evident in the deterioration of the muscular, skeletal, gastrointestinal, and nervous systems. For instance, the loss of muscle mass, strength, and function is a well-recognized hallmark of aging [2]. Emerging evidence highlights the significance of the gut–muscle axis, a bidirectional communication pathway through which gut health directly influences muscle physiology [3]. Intestinal aging, marked by impaired barrier integrity, reduced motility, microbial dysbiosis, and chronic inflammation, has been implicated in exacerbating muscle aging. Chronic inflammation plays a central role in both intestinal and muscular degeneration. Inflammatory conditions compromise the integrity of epithelial tight junctions, leading to increased intestinal permeability [4]. At the same time, macrophages undergo functional changes, secreting large amounts of pro-inflammatory cytokines such as TNF-α, IL-1β and IL-6, which further exacerbate the inflammatory response [4]. Oxidative damage caused by reactive oxygen species (ROS) or free radicals is another key factor that accelerates aging. ROS can lead to lipid peroxidation, enzyme dysfunction, and DNA damage, thereby impairing normal gut and muscle functions [5]. Therefore, targeting inflammation and oxidative stress is vital for preserving gut health and preventing age-associated intestinal and muscle degeneration.

Fermentation is widely utilized to enhance the nutritional quality and bioactivity of food ingredients [6,7]. A small black soybean fermented by *Bifidobacterium animalis* alleviated muscle mass decrease and improved the expression of biomarkers including total MHC, myf6, phosphorylated AKT and Tfam, which are related to myoblast differentiation, muscle protein synthesis, and mitochondrial generation in the muscle [8]. *Lactiplantibacillus plantarum*-fermented kiwifruit juice has higher phenolic and flavonoid content, enhancing its antioxidant capacity. It reduced LPS-induced colonic malondialdehyde (MDA) levels and increased the activities of glutathione peroxidase (GSH-Px), superoxide dismutase (SOD), and catalase (CAT), thereby mitigating oxidative stress in the gut [9]. Fermentation of goji berry juice with *Lactobacillus casei*, *Lactobacillus plantarum*, and *Lactobacillus rhamnosus* reduced IL-1β, IL-6, and TNF-α levels to alleviate inflammation while increasing tight junction proteins (ZO-1, Claudin-1, Occludin) to protect the intestinal barrier [10]. Collectively, these findings suggested that fermenting plant-based materials may represent a promising approach for improving muscle and intestinal health through antioxidant and anti-inflammatory mechanisms.

Lychee (Litchi chinensis) is rich in various bioactive compounds, particularly polyphenols such as proanthocyanidins, catechins, and gallic acid. Lychee has been traditionally used as an analgesic in ancient Chinese medicine [11,12]. Fermentation of lychee peel by *Aspergillus awamor* enhanced the production of catechin and quercetin, leading to increased DPPH radical scavenging activity [13]. Lychee fermented with *Lactiplantibacillus plantarum* HU-C2W showed significant increases in γ-aminobutyric acid and flavonoid content, thereby enhancing hepatic total antioxidant capacity while reducing malondialdehyde levels to alleviate oxidative damage [14]. Fermentation of lychee with *Lacticaseibacillus casei* CICC 6117 significantly increased its total phenolics, total flavonoids, and exopolysaccharides, exhibiting protective effects against cyclophosphamide-induced damage to immune organs by enhancing antioxidant capacity [15]. These findings suggested that lychee fermentate may also exert protective effects on muscle and intestinal health [16].

Furthermore, mixed-strain fermentation has been reported to be more effective than single-strain fermentation in promoting the release of phenolic compounds from lychee pulp and may yield enhanced biological activities [17,18,19]. During the fermentation process, yeasts catalyze the conversion of glucose and fructose into ethanol and carbon dioxide while simultaneously producing vitamins, amino acids, and other beneficial metabolites. Lactic acid bacteria utilize carbon sources and amino acids to generate a variety of bioactive substances, including organic acids, peptides, fatty acids, polysaccharides, and vitamins. *Acetobacter* strains oxidize alcohol to acetic acid and contribute to the production of antioxidant components such as polyphenols [20]. Therefore, this study employed a mixed-culture fermentation approach using *Lactiplantibacillus plantarum*, *Lacticaseibacillus casei*, *Saccharomyces cerevisiae*, and *Acetobacter pasteurianus* SP021 to produce a lychee fermentate. Using a D-galactose-induced murine model, we systematically investigated its effects on alleviating intestinal and muscle aging. Moreover, untargeted metabolomic analysis was conducted to identify and characterize the key bioactive constituents potentially responsible for its health-promoting effects.

## 2. Materials and Methods

### 2.1. Materials and Main Reagents

Lychee fruits (cv. Heiye) were purchased from the local market. *Lactiplantibacillus plantarum* and *Lacticaseibacillus casei* were purchased from Shanghai Jiaoda Onlly Co., Ltd., China (Shanghai, China). *Saccharomyces cerevisiae* used in this study was instant dry yeast, which was purchased from Yinglian Food Supplements Co., Ltd., China (Shanghai, China). *Acetobacter pasteurianus* SP021 was purchased from the China General Microbiological Culture Collection Center (CGMCC) (Beijing, China), with the collection number 25794. D-galactose was purchased from Sigma-Aldrich (St. Louis, MO, USA). A Hematoxylin and Eosin (H&E) staining kit and Alcian Blue Periodic Acid Schiff (AB-PAS) staining kit were purchased from Solarbio (Beijing, China). Trizol reagent was purchased from Cwbio (Taizhou, China). A Qubit RNA BR Assay Kit was acquired from Invitrogen (Eugene, OR, USA). HiScript III RT SuperMix and ChamQ SYBR Master Mix were purchased from Accurate Biology (Changsha, China). Primary antibody against MUC2 or ZO-1 was purchased from Proteintech (Wuhan, China). Primary antibody against Claudin-1 or Occludin and secondary antibody conjugated with Alexa Fluor 488 was purchased from Abcam (Shanghai, China). A D-lactic acid assay kit and GSH-Px assay kit were purchased from Jiancheng Bioengineering Institute (Nanjing, China). An MDA assay kit and SOD assay kit were purchased from Solarbio (Beijing, China).

### 2.2. Lychee Fermentate Preparation

Fresh lychee fruits were washed thoroughly and pressed to obtain lychee juice. The fermentation process followed a patented two-stage protocol (ZL202210565739.1). Firstly, lychee juice (including pulp) was supplemented with sucrose to a final concentration of 0.15% (*w*/*v*). The juice was then inoculated with 0.05% (*w*/*w*) mixture containing *S. cerevisiae*, *L. plantarum* and *L. casei*, to reach a final concentration of 5 × 10^5^ CFU/mL, 4 × 10^5^ CFU/mL, and 4 × 10^5^ CFU/mL, respectively. The first stage of fermentation was conducted at 30 °C for 72 h. Secondly, the fermented liquid was filtered, and the filtrate underwent secondary fermentation with 0.3% (*w*/*v*) *A. pasteurianus* SP021 at 30 °C for an additional 72 h. Finally, the product was sterilized by pasteurization (80 °C for 30 min) to obtain LF. Lychee juice (LJ) processed under identical pasteurization conditions served as a control.

### 2.3. Animal and Experimental Design

Six-week-old male C57BL/6J mice (Vital River Laboratory Animal Technology Co., Ltd., Beijing, China) were acclimatized for one week under standardized environmental conditions (24 ± 2 °C, 50–60% humidity, 12-h light/dark cycle). Following acclimatization, the mice were randomly grouped as the control group (*n* = 6), model group (*n* = 6), low-dose lychee fermentate group (LLF, 26.1 mg/kg·bw, *n* = 6), and high-dose lychee fermentate group (HLF, 78.3 mg/kg·bw, *n* = 6). The dose selection of lychee fermentate was based primarily on the effective gavage concentration reported by Huang et al., who demonstrated that lychee fermentate alleviated DSS-induced colitis [16]. The control group received daily intraperitoneal and oral administrations of sterile saline. The other groups were administered D-galactose intraperitoneally (500 mg/kg·bw/day) and lychee fermentate orally at the indicated doses for eight consecutive weeks. At the end of the intervention period, fecal samples were collected for gut microbiota analysis. After a 12-h fast, mice were weighed, anesthetized with diethyl ether, and blood and colon samples were collected. These samples were immediately stored at −80 °C.

### 2.4. Muscle Strength and Endurance Analysis

Grip strength was assessed using an electronic grip strength meter (Beijing Zhongshidichuang Science and Technology Development Co., Ltd., Beijing, China). Each mouse was gently allowed to grasp a horizontal metal bar with its forepaws while the investigator applied a steady pull to the tail in the opposite direction, keeping the body in a horizontal position. The peak force (g) exerted by the mouse before release was recorded. Each mouse was tested three times, and the average value was used for subsequent analysis. Suspension endurance was measured using a wire mesh grid (43 cm × 43 cm; mesh size: 12 mm × 12 mm; wire diameter: 1 mm) mounted on a 4 cm wooden frame. Mice were placed individually in the center of the grid, which was then inverted 180° and suspended 40 cm above a soft pad. The latency to fall was recorded as the endurance time. Exercise endurance was conducted using the animal experimental treadmill (Beijing zhongshidichuang Science and technology development Co., Ltd., Beijing, China) equipped with an electric stimulation module set to 0.3 mA. Prior to testing, mice were acclimated through progressive training once daily for two consecutive days, with each session lasting 10 min. The acclimation protocol consisted of three speed stages: 5 m/min for 3 min, 10 m/min for 3 min, and 15 m/min for 2 min. During the formal test, the treadmill protocol included the following stages: the initial speed is 5 m/min (lasting for 1 min), the first-level speed is 10 m/min (lasting for 2 min), and the second-level speed is 20 m/min (lasting until exhaustion). Exhaustion was defined as the inability of a mouse to continue running for 10 consecutive seconds despite electric stimulation. The total running time until exhaustion was recorded as the endurance time.

### 2.5. Histopathological Analysis

The proximal colon tissue samples and gastrocnemius muscle samples were excised and immediately immersed in 4% paraformaldehyde for fixation at room temperature for at least 48 h. After fixation, the tissues were sequentially dehydrated in an ascending series of ethanol concentrations, rendered transparent in xylene, and subsequently embedded in paraffin. Paraffin sections were then prepared, deparaffinized, and rehydrated using standard protocols. Finally, the sections were stained with a Hematoxylin and Eosin (H&E) staining kit and a Alcian Blue Periodic Acid Schiff (AB-PAS) staining kit, following the manufacturer’s instructions. The stained slides were examined under an optical microscope (CTR6, Leica, Germany) for histopathological analysis. Muscle fiber cross-sectional area was measured using Image J software (v1.8.0) on stained tissue sections.

### 2.6. Transcriptional Analysis by Real-Time qPCR

Total RNA from 30 mg colonic tissue sample was extracted by Trizol reagent according to the manufacturer’s instructions. The RNA quantity was then accurately assessed by the Qubit RNA BR Assay Kit. Subsequently, 1 g of total RNA was used to generate the complementary DNA by HiScript III RT SuperMix. Quantitative real-time polymerase chain reaction (RT-qPCR) was then performed using ChamQ SYBR Master Mix with Step One Plus Real-time PCR System. Primers are listed in Appendix A. The relative quantification of the target gene was calculated by comparing with the β-actin as reference according to the 2^−ΔΔCT^ method [21]. To further evaluate intestinal aging at the gene level, the mRNA expression levels of p16 and p21, which are two well-established markers of cellular senescence, were quantified in colonic tissues.

### 2.7. Immunofluorescence Analysis

Colon tissue sections were incubated with a primary antibody against MUC2, followed by a secondary antibody conjugated with Alexa Fluor 488. Three additional sets of tissue sections were immunostained with primary antibodies against ZO-1, Claudin-1 and Occludin, using the same secondary antibody. Immunolabeled sections were examined under a fluorescence microscope for visualization of target protein expression.

### 2.8. Determination of D-Lactate, MDA and Antioxidant Enzymes

The levels of D-lactate, GSH-PX, MDA and SOD in serum were detected by D-lactate assay kit, GSH-PX assay kit, MDA assay kit and SOD assay kit according to the manufacturer’s instructions, respectively.

### 2.9. Microbial Community Analysis

Total microbial DNA was extracted from mouse feces using the CTAB/SDS method and examined by electrophoresis on a 1% agarose gel. The V3-V4 regions of the bacterial 16S rRNA gene were amplified using primers 341F (5′-CCTAYGGGRBGCASCAG-3′) and 806R (5′-GGACTACNNGGGTATCTAAT-3′). Following amplification, PCR products were purified, and sequencing was performed by Majorbio Bio-Pharm Technology Co., Ltd. (Shanghai, China) using an Illumina MiSeq platform with a paired-end sequencing strategy. The sequencing reads were overlapped and spliced to obtain high-quality sequences. Amplicon sequence variants (ASVs) were then clustered, and species taxonomy was annotated. The Alpha diversity of microbiota was evaluated with the Shannon index. Beta diversity was assessed by hierarchical clustering tree and principal coordinate analysis (PCoA) based on weighted unifrac distance. The differentially abundant taxa were identified by Tukey and Kruskal–Wallis H tests with *p* < 0.05.

### 2.10. Untargeted Metabolomics

The metabolites of LF were detected by untargeted metabolomics using an UHPLC (1290 Infinity LC, Agilent Technologies, Santa Clara, CA, USA) coupled to an AB Sciex TripleTOF 6600 quadrupole time-of-flight mass spectrometer. Samples were separated on a C-18 column at 40 °C with a flow rate of 0.4 mL/min and an injection volume of 2 μL. The mobile phase consisted of 25 mM ammonium acetate and 0.5% formic acid in water (phase A) and methanol (phase B), respectively. The gradient elution was as follows: 0–0.5 min, 5% B; 0.5–10 min, 5–100% B; 10–12 min, 100% B; 12–12.1 min, 100% to 5% B; and 12.1–16 min, 5% B. Samples were kept in an automatic sampler at 4 °C. ESI source conditions were as follows: Ion Source Gas1 and Gas2 at 60, curtain gas at 30, source temperature at 600 °C, and IonSpray Voltage Floating (ISVF) at ±5500 V. MS acquisition was performed in a range of 60–1000 Da, with a TOF MS scan accumulation time of 0.20 s/spectrum. For auto MS/MS acquisition, the range was 25–1000 Da, and the accumulation time for product ion scan was set at 0.05 s/spectrum. Data were acquired using information-dependent acquisition (IDA) in high-sensitivity mode. For data processing, raw MS data were converted to MzXML files using ProteoWizard MSConvert and imported into XCMS software (XCMS version 4.0). Peak picking and grouping were performed with specific parameters. CAMERA was used for isotope and adduct annotation, and only variables with more than 50% non-zero values in at least one group were retained. Metabolite identification was performed by matching accurate *m*/*z* values (<10 ppm) and MS/MS spectra against an in-house database of authentic standards.

### 2.11. Statistical Analysis

All data were analyzed with GraphPad Prism 9.5 software (San Diego, CA, USA) and expressed as mean ± standard deviation (SD). Univariate analysis of variance (ANOVA) was used to assess statistical significance between groups. *p* < 0.05 was considered to be statistically significant.

## 3. Results

### 3.1. The Effects of LF on Skeletal Muscle Function

The experimental design used to assess the muscle protective effect of LF on Dgalactose-induced aging in C57BL/6J mice was illustrated in Figure 1A. Neither D-galactose injection nor LF supplementation significantly affected the body weight gain (Figure 1B) and daily food intake of mice in each group (Appendix A). Compared to the control group, D-galactose-treated mice showed a significant decrease in grip strength (*p* < 0.05; Figure 1C). In contrast, HLF maintained grip strength at the level of the control group. Suspension time, exhaustion time, and indicators of muscle endurance followed similar trends (Figure 1D,E). The results indicated that lychee fermentate could restore muscle endurance caused by D-galactose.

H&E staining showed that muscle fibers in the control group were polygonal and neatly arranged (Figure 1G). In the model group, muscle fibers appeared disorganized with markedly widened inter-fiber spaces. The HLF group exhibited substantial restoration of fiber organization. Quantitative analysis of muscle fiber cross-sectional area (CSA) revealed a significant reduction in the D-galactose group (Figure 1F). CSA was significantly improved in the HLF group (*p* < 0.05). These findings suggested that LF alleviated D-galactose-induced muscle loss in mice.

### 3.2. LF Attenuated Intestinal Aging and Restored Intestinal Barrier Function

Based on the gut–muscle axis theory, there is a close relationship between muscle health and intestinal aging [22]. The improvement of muscle function in D-galactose-induced mice by LF is likely mediated through the restoration of intestinal function. Following 8 weeks of D-galactose administration, the intestinal transit capacity of mice was significantly impaired (Appendix A). The total intestinal transit time in the D-gal group (144.0 ± 18.8 min) was markedly prolonged compared to the control group (107.3 ± 22.6 min). The LLF and HLF groups exhibited shortened transit times of 117.6 ± 26.2 min and 104.1 ± 14.6 min, respectively, with the HLF group showing a significant improvement relative to the D-gal group (Figure 2A, *p* < 0.05). Furthermore, the transcription of both p16 and p21 was significantly upregulated by D-galactose treatment compared to the control group. In contrast, LF supplementation led to a significant downregulation of these senescence associated genes relative to the D-gal group (*p* < 0.05) (Figure 2B,C), indicating a protective effect against D-gal-induced intestinal cellular aging.

H&E staining showed that D-gal led to significant colonic tissue damage, characterized by crypt distortion and inflammatory infiltration (Figure 2E). Intervention with LF significantly ameliorated these colonic tissue damage. AB-PAS staining showed that the Con group had an average of 35.1 PAS^+^ cells per crypt, whereas the D-gal group had only 24.3 (Figure 2E). Notably, HLF treatment significantly increased the number of PAS^+^ goblet cells to 33.7 per crypt (Figure 2F). Furthermore, D-gal significantly reduced the transcriptional level of MUC2 in colon tissue (*p* < 0.05). HLF supplementation effectively restored MUC2 expression to the levels of the control group (Figure 2G,H). Immunofluorescence staining further confirmed these results at the protein level (Figure 2G,I). Therefore, HLF mitigated D-gal-induced colonic mucosal injury by preserving goblet cell populations and promoting MUC2-mediated mucus secretion, thereby contributing to the maintenance of intestinal epithelial integrity.

In addition, D-gal significantly increased D-lactic acid concentration in serum, indicating increased intestinal permeability (Figure 2D). Compared with the model group, LF intervention at high doses reduced the D-lactic acid levels. Moreover, RT-qPCR results showed that HLF treatment significantly upregulated the expression of *claudin-1*, *occludin* and *ZO-1* genes, restoring their levels close to those observed in the control group (Appendix A), which was further confirmed by colon immunofluorescence staining analysis (Appendix A). These findings suggested that HLF effectively mitigated D-gal-induced disruption of intestinal barrier function by upregulating the expression of the key TJ proteins.

### 3.3. LF Attenuated Oxidative Stress and Inflammation

In terms of inflammation levels and antioxidant capacity, the expression of proinflammatory factors in the D-Gal group, including IL-1β, IL-6, TNF-α, CXCL1 and MCP1, were significantly increased by 2.50-, 2.57-, 2.58-, 3.88- and 2.99-fold, respectively. Notably, HLF almost restored inflammation levels to the level of the control group (Figure 3A–E). Furthermore, LF intervention significantly decreased the levels of the oxidative marker MDA. The MDA concentration in the HLF group was reduced to 24.650 ± 3.84 nmol/mL, whereas that of the D-gal group was 31.4 ± 3.36 nmol/mL. In addition, HLF treatment significantly increased the activities of the antioxidant enzymes GSH-PX and SOD (Figure 3F–H). These findings suggested that HLF effectively mitigated D-gal-induced oxidative stress and inflammation in mice.

### 3.4. The Modulatory Effects of LF on Gut Microbiota

To investigate the impact of LF on gut microbiota alterations induced by D-galactose, 16S rRNA gene sequencing was performed on fecal samples. A total of 575,359 high-quality reads corresponding to 10,421 ASVs at single-nucleotide resolution were obtained, with an average of 31,964 reads per sample, which were suitable for downstream analysis of the gut microbial community (Appendix A). PCoA analysis at the ASV level revealed distinct clustering in beta diversity among the three groups (Figure 4A). HLF group exhibited a microbial composition more similar to the control group than to the model group, indicating that LF intervention partially restored the gut microbial community structure disrupted by D-galactose (Figure 4B). The Shannon index showed that the HLF group was significantly higher than the D-gal group (*p* < 0.05), suggesting that LF can upregulate the diversity of gut microbiota.

The Circos plot revealed that the four most abundant bacterial phyla in the gut microbiota were Bacteroidota, Firmicutes, Actinobacteriota, and Campilobacterota (Figure 4C). D-gal increased the abundance of Bacteroidota while decreasing Firmicutes (*p* < 0.05). HLF intervention reversed these alterations, increasing the relative abundance of Firmicutes and decreasing Bacteroidetes, bringing their proportions closer to the control group levels (Figure 4D). At the genus level, D-gal treatment increased the relative abundance of *Muribaculaceae*, which was associated with pro-inflammatory responses. Conversely, HLF supplementation significantly decreased the abundance of *Muribaculaceae* and increased the relative abundances of beneficial genera, including *Clostridia*_UCG-014, *Lachnospiraceae*_NK4A136_group, *Alistipes*, *Ruminococcaceae* and *Colidextribacter* (Figure 4E,F).

Spearman correlation analysis was conducted to explore the relationship between specific microbial taxa and host physiological parameters. Pro-inflammatory factors including IL-1β, IL-6, TNF-α, CXCL1 and MCP1 were positively correlated with *Muribaculaceae* but negatively correlated with *Clostridia*_UCG-014, *Alistipes*, and *Ruminococcaceae*, suggesting that these beneficial bacteria may exert anti-inflammatory effects (Figure 4G). Muscle strength and endurance were shown to be positively correlated with *norank_o__Clostridia*_UCG-014, *Alistipes* and *norank_f__Ruminococcaceae*. Furthermore, gut barrier-related markers including MUC2, ZO-1, Claudin-1 and Occludin were negatively correlated with *Muribaculaceae*, but positively correlated with *Ruminococcaceae* and *Clostridia*_UCG-014. These findings indicated that LF might attenuate D-galactose-induced skeletal muscle and intestinal injury in mice by modulating the gut microbiota composition.

### 3.5. Phenolic Acids and Flavonoids Were Enriched by Fermentation

The effects of fermentation on metabolites in lychee juice were uncovered through untargeted metabolomics. Through principal component analysis (PCA), the cluster of LJ and LF samples can be completely separated under the positive (ESI+) and negative (ESI−) ion modes, respectively (Figure 5A). OPLS-DA models could clearly distinguish the LJ and LF samples with Q2cum of 0.991 and 0.994 in ESI+ and ESI− modes, respectively (Figure 5B). A total of 976 metabolites in the LJ and LF samples were identified. Differential metabolites in pairwise comparison between the LJ and LF groups were selected by combining variable importance in projection (VIP) ≥ 1, |Fold change (FC)| ≥ 1.5 and *p* value < 0.05. There were 162 differential metabolites after volcano plot analysis, of which 134 were upregulated and 28 were downregulated after the fermentation process (Figure 5C). KEGG enrichment analysis showed that the predominant differential metabolites in the LF group were mainly enriched into several metabolic pathways associated with different functions, such as amino acid metabolism, vitamin metabolism, flavonoid metabolism, aminobenzoate degradation, etc. (Appendix A).

The representative differential metabolites in the LF vs. LJ group are listed in Appendix A. Furthermore, a heatmap was employed to reveal the differences in the relative content of some selected differential metabolites in each group (Figure 5D). The fermentation process markedly depleted primary metabolites, specifically carbohydrates and amino acids, highlighting their utilization by microorganisms as substrates for growth and secondary metabolism. The content of glucose, fructose and sucrose in LF group were significantly down-regulated, with a decrease of 8.94, 7.79 and 82.97-fold, respectively. In contrast to the depletion of primary metabolites, a broad spectrum of secondary metabolites, particularly phenolic acids and flavonoids, were significantly enriched following fermentation process. This microbial enzymatic biotransformation is presumed to involve the hydrolysis of glycosidic bonds and subsequent deglycosylation of conjugated phenolics, thereby releasing aglycone forms with increased bioavailability and biological activity. Notably, several flavonoids and their derivatives were increased in the LF group (Figure 5E), including eriodictyol (7.39-fold), taxifolin (3.43-fold), and Isorhamnetin (11.78-fold) and luteolin (1.50-fold), all of which are recognized for their potential antioxidant and anti-inflammatory properties. Meanwhile, phenolic acid and its derivatives were also significantly increased in the LF group, with ethyl caffeate, vanillic acid, gallic acid, 4-hydroxycinnamic acid and caffeic acid increasing by 30.76, 6.89, 4.67, 2.84 and 2.50-fold, respectively. Targeted metabolomics analysis further verified that the content of gallic acid in the LF group reached 155.74 ± 14.65 ng/mL after fermentation, which significantly increased by 5.23-fold. The concentration of vanillic acid in the LF group reached 303.40 ± 19.80 ng/mL, which was 10.82-fold higher than that of the LJ group. These shifts suggested activation of microbial pathways related to phenylpropanoid metabolism and aromatic compound degradation. In addition to polyphenols, the microbial aromatic metabolite L-(–)-3-phenyllactic acid and 2-isopropylmalic acid were enriched by 172.80-fold and 121.08-fold, respectively. The substantial accumulation of these metabolites suggests the involvement of microbial hydrolytic and biosynthetic activities in transforming endogenous lychee constituents into relevant secondary metabolites. These metabolomic changes underline microbial capability in modifying lychee constituents, enhancing functional compounds with potential antioxidant, anti-inflammatory, and health-promoting properties.

## 4. Discussion

In this study, LF showed significant protective effects against D-galactose-induced muscle dysfunction, intestinal aging, oxidative stress, inflammation, gut barrier dysfunction and gut microbiota dysbiosis in mice (Figure 6). These findings indicated that the fermentation of lychee yielded a potent cocktail of bioactive metabolites through microbial transformations, which was further confirmed by untargeted metabolomics. During fermentation, lactic acid bacteria and yeast produced a suite of enzymes capable of degrading lychee’s cell wall components, resulting in the liberation of various bioactive compounds. In lychee fermentate, the liberation of polyphenolic compounds such as kaempferol-3-glucoside-3″-rhamnoside, cyanidin 3-o-glucoside, 4-hydroxybenzoic acid 4-o-glucoside, and luteolin 7-o-rutinoside from their cell wall-bound forms were facilitated by microbial enzymatic activities (Appendix A). Phenolic acids formerly conjugated to cell-wall components become free, such as p-coumaric acid increased 3.7-fold and ethyl caffeate rose 30-fold postfermentation. Additionally, fermentation yielded a 4.7-fold increase in gallic acid, reflecting tannase-mediated hydrolysis of gallotannins to release free gallic acid. Such enzymatic deesterification of “insoluble” phenolics have been well-documented to elevate free phenolic content and bioavailability in fermented foods [23].

Lactic acid bacteria and other fermentative microbes express β-glycosidases that hydrolyze glycosidic bonds on polyphenols, removing sugar moieties to release aglycone forms [24]. In LF, this deglycosylation is reflected by depletion of numerous flavonoid glycosides alongside a rise in their aglycones. For example, isorhamnetin-3-galactoside was markedly reduced after fermentation, while free isorhamnetin aglycone increased 12-fold (Appendix A and Figure 5E). Likewise, aglycones such as eriodictyol (7.4-fold) and taxifolin (3.4-fold) accumulated in LF, presumably liberated from their conjugated precursors by glycosidases. Consistent with previous studies, fermentations enriched in flavonol aglycones (kaempferol, isorhamnetin, naringenin, etc.) have been shown that significantly increased total phenolic content and antioxidant activity of plant-based fermented products [25]. Microbial transformation of phenolic acid and its derivatives was another hallmark of lychee fermentation. For instance, 2,5-dihydroxybenzoic acid, gallic acid, ethyl ferulate and caffeic acid increased 6.19, 4.67, 3.99 and 2.50-fold in the lychee fermentate, respectively. The formation of these compounds likely proceeds via microbial hydroxylation of cinnamic or p-coumaric acid (yielding caffeic acid) and subsequent cleavage of side chains to form corresponding benzoic acid and its derivatives.

Fermentation also introduced metabolites that were absent in fresh lychee, indicating de novo microbial synthesis or unique amino acid conversions. One prominent example is indole-3-lactic acid (ILA), which was undetectable in unfermented juice but accumulated in LF. ILA is exclusively produced via microbial tryptophan catabolism; indeed, *L. plantarum* and other lactic acid bacteria are known to convert tryptophan to ILA [26]. Another fermentation-derived compound, 2-isopropylmalic acid (2-IPMA), increased 121-fold in LF. 2-IPMA is an intermediate of leucine biosynthesis that certain microbes excrete. Notably, *Saccharomyces cerevisiae* has been reported to excrete 2-IPMA during growth. In addition to 2-IPMA, other metabolites such as citramalic acid and kojic acid were exclusively detected in LF, indicating their de novo synthesis by fermentative microbes. These microbial-specific metabolites highlighted how the fermentation microbes (lactic acid bacteria, yeast, and *Acetobacter*) can generate novel metabolites.

Flavonoids and phenolic acids generated by fermentation are widely recognized for their potent anti-inflammatory and antioxidant properties, which contribute to the attenuation of skeletal muscle and intestinal aging [27,28,29]. Among these, compounds such as isorhamnetin, eriodictyol, syringic acid, and quercetin have exhibited significant efficacy in alleviating intestinal injury or muscle atrophy [30,31,32,33,34,35,36]. Notably, isorhamnetin could enhance nuclear factor erythroid 2-related factor 2 (Nrf2) activity, leading to increased expression of heme oxygenase-1 (HO-1) and other antioxidant enzymes, contributing to its protective effects against intestinal inflammation [32]. Additionally, isorhamnetin inhibits TNF-α-induced apoptosis in HUVECs by reducing NF-κB expression [37]. Furthermore, gallic acid can mitigate pentachlorophenol-induced intestinal damage through its antioxidant activity [38], while p-hydroxybenzoic acid has been shown to enhance the intestinal barrier function in mice [39].

In addition, lychee fermentate was rich in a variety of microbial-specific metabolites with anti-inflammatory properties. For instance, indole-3-lactic acid, which significantly increases in content after fermentate, has been widely recognized for its ability to alleviate intestinal inflammation by inhibiting the expression of pro-inflammatory cytokines such as TNF-α [40]. This effect is mediated through the activation of the aryl hydrocarbon receptor (AhR) and Nrf2 pathways, which are crucial for maintaining intestinal homeostasis and mitigating oxidative stress. Specifically, ILA has been demonstrated to upregulate AhR-target genes like *CYP1A1* and Nrf2-targeted genes such as *GPX2* and *SOD2*, contributing to its protective effects against inflammation and intestinal senescence [40,41]. The combined action of these bioactive compounds underscored the potential of fermented lychee as a functional food with anti-inflammatory benefits.

Lastly, the modulation of gut microbiota by lychee fermentate was a pivotal mechanism linking these metabolites to gut health. Diet-derived polyphenols and fermentation metabolites often act as prebiotics or microbial modulators, selectively enriching beneficial microbes while inhibiting opportunistic pathogens [42]. Consistent with prior research, our findings revealed that D-gal-induced intestinal injury in mice resulted in decreased gut microbiota diversity, an elevated abundance of Bacteroidetes, and a reduced abundance of Firmicutes [43]. The relative abundance on phylum level in the HLF group was significantly increased in Firmicutes and decreased in Bacteroidetes compared to the D-gal group, suggesting that LF tended to regulate gut microbiota homeostasis by increasing Firmicutes and inhibiting Bacteroidetes. Additionally, the abundance of *Clostridia*_UCG-014 and *Alistipes* in the HLF group was significantly higher than that in the D-gal group. *Clostridia*_UCG-014 is reported to be decreased in DSS-induced colitis mice, and its increased abundance is closely associated with inhibition of the development of rheumatoid arthritis, while *Alistipes* can fight against inflammation caused by a high-fat diet [44,45]. KEGG pathway enrichment analysis further indicated that D-galactose disrupted microbial metabolic pathways associated with energy and lipid metabolism while upregulating disease-related pathways. High-dose lychee fermentate (HLF) reversed these alterations, particularly downregulating aging- and inflammation-related pathways and restoring metabolic homeostasis, suggesting a protective modulation of gut microbiota function against aging-associated dysbiosis. (Appendix A). This suggests that LF may modulate microbial functions linked to intestinal aging, potentially contributing to the maintenance of gut health during the aging process.

## 5. Conclusions

This study demonstrates that lychee fermentate (LF) significantly alleviates D-galactose-induced muscle dysfunction, intestinal aging, oxidative stress, inflammation, and gut microbiota dysbiosis in mice. The protective effects of LF might be attributed to the microbial transformation of lychee during fermentation, which generates bioactive metabolites with potent antioxidant and anti-inflammatory properties. Untargeted metabolomics analysis confirmed the enrichment of phenolic acids and flavonoids in LF, which play a crucial role in restoring gut health and muscle function. Furthermore, LF modulated gut microbiota composition, restoring the relative abundance of Firmicutes and Bacteroidetes, and enriching beneficial microbial taxa, which likely contribute to its therapeutic effects. These findings suggest that lychee fermentate holds promise as a functional food for mitigating age-related physiological decline, particularly through its ability to combat oxidative stress, inflammation, and microbiota imbalance, ultimately supporting healthy aging.

The primary limitation of this study is the absence of a “D-galactose + unfermented lychee juice” control group in the animal experiments, as unfermented lychee juice was included only for metabolomic analyses. This design prevents definitive attribution of the observed benefits—such as enhanced antioxidant activity and modulation of gut microbiota—to fermentation-derived metabolites versus the intrinsic bioactive compounds present in lychee juice, including phenolics and polysaccharides. Consequently, future studies directly comparing fermented and unfermented lychee with this mouse model are warranted.

## Figures and Tables

**Figure 1 foods-14-03684-f001:**
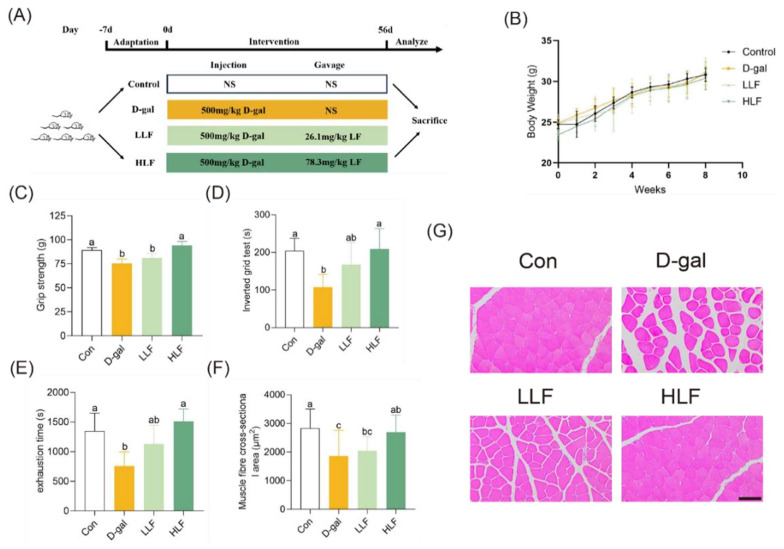
Lychee fermentate alleviated D-galactose-induced skeletal muscle aging in mice. (**A**) Schematic diagram illustrating the experimental design. (**B**) Body weight measurements of mice across different treatment groups. (**C**–**E**) Assessment of physical performance using all-limb force, longest suspension time, and time to exhaustion evaluated by handgrip, hanging wire tests, and treadmill, respectively. (**F**) Cross-sectional area of skeletal muscle fibers. (**G**) Representative H&E staining of the gastrocnemius muscle. Scale bars = 100 μm. Data are presented as mean ± standard deviation (SD). The letters above the bars indicate statistical differences (*p* < 0.05).

**Figure 2 foods-14-03684-f002:**
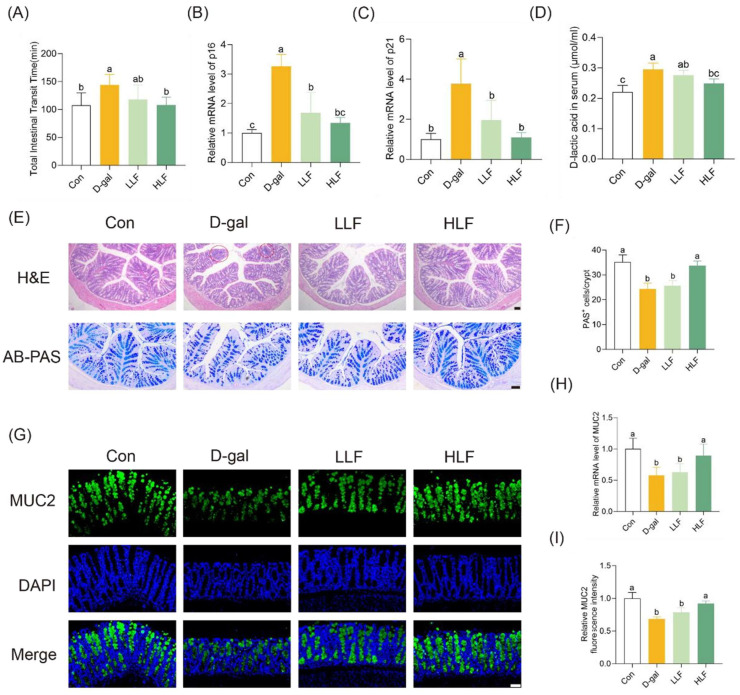
Lychee fermentate ameliorated D-galactose-induced colonic morphological alterations in mice. (**A**) Total intestinal transit time of different groups. (**B**,**C**) The mRNA levels of *p16* and *p21* genes in colon tissues, respectively. (**D**) The serum levels of D-lactic after treatment with LF. (**E**) Representative H&E and AB-PAS staining of the colon. (**F**) Quantification of PAS^+^ goblet cells per crypt (*n* = 6). (**G**) Immunofluorescence staining of MUC2 (green) in colonic sections. (**H**) Relative mRNA expression levels of MUC2 in colon tissues. (**I**) Quantitative analysis of MUC2 fluorescence intensity from immunofluorescence images. Scale bars = 100 μm. The letters above the bars indicate statistical differences (*p* < 0.05).

**Figure 3 foods-14-03684-f003:**
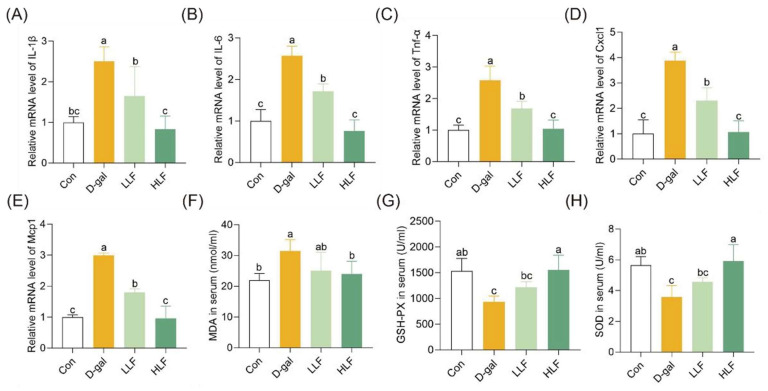
Lychee fermentate attenuated inflammation and oxidative stress in D-galactose-induced intestinal injury in mice. (**A**–**E**) Relative mRNA expression levels of pro-inflammatory cytokines (IL-1β, IL-6, TNF-α, CXCL1, and MCP1) in colon tissues. (**F**–**H**) Effects of LF intervention on MDA, GSH-PX and SOD in serum. Data are presented as mean ± standard deviation (SD). The letters above the bars indicate statistical differences (*p* < 0.05).

**Figure 4 foods-14-03684-f004:**
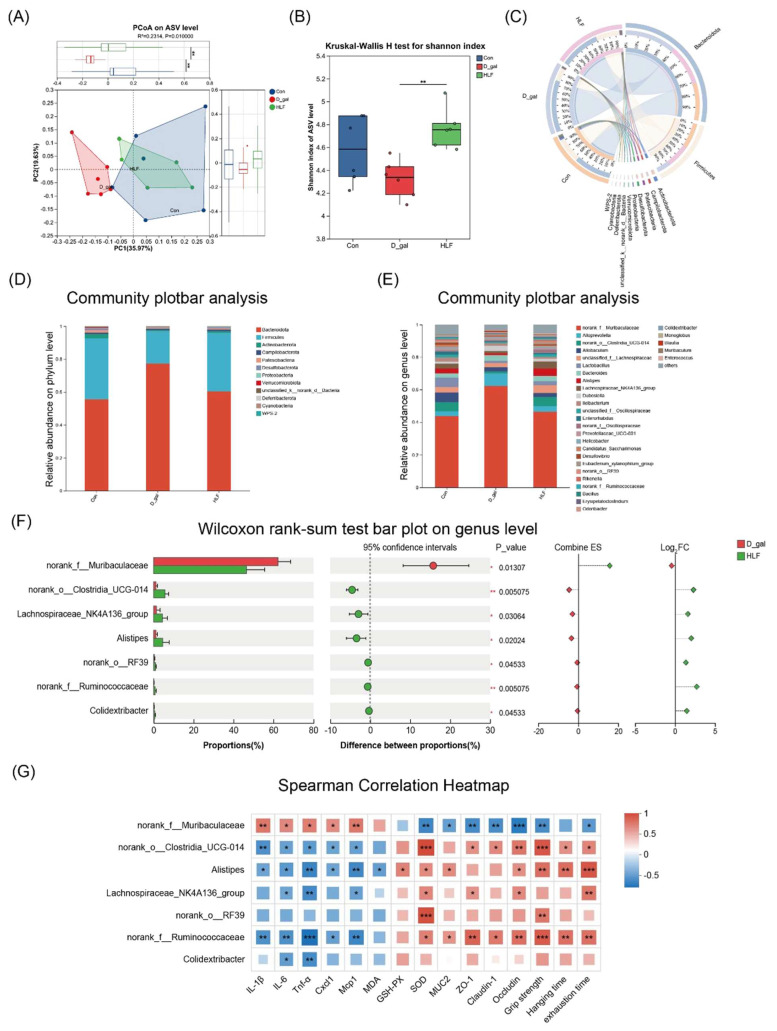
Lychee fermentate modulated gut microbiota composition in D-galactose-induced intestinal injury in mice. (**A**) Principal coordinate analysis (PCoA) plot on ASV level of microbiota. (**B**) Shannon index representing α-diversity of gut microbiota across the three groups. (**C**) Circos plot visualization of bacterial phylum horizontal abundance. (**D**,**E**) Phylum level and genus level of intestinal flora. (**F**) Differential abundance analysis at the genus level between D-gal group and HLF group. (**G**) Spearman’s correlation analysis between bacterial at the genus level and metabolic phenotypes. * *p* < 0.05, ** *p* < 0.01, *** *p *< 0.001.

**Figure 5 foods-14-03684-f005:**
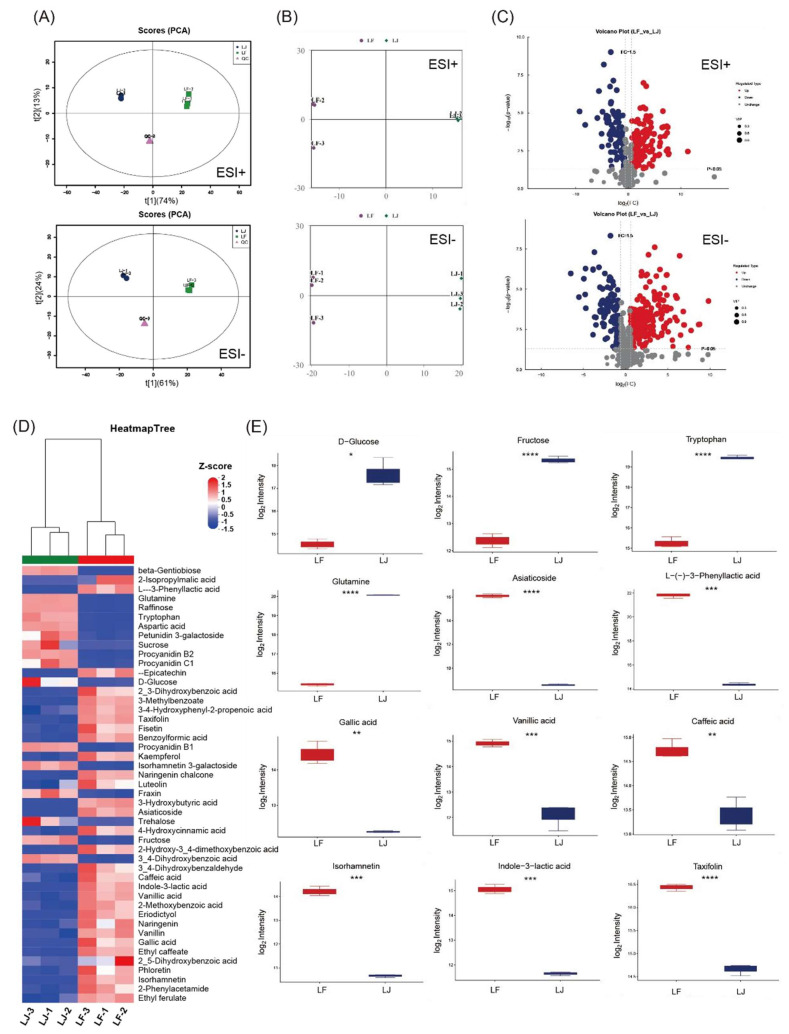
Untargeted metabolomics analysis of lychee juice and lychee fermentate. (**A**) PCA of lychee juice and lychee fermentate in ESI+ and ESI− modes. (**B**) OPLS-DA score plots of lychee juice and lychee fermentate. (**C**) Volcano plots of differential metabolites between LJ and LF. (**D**) Heatmaps and (**E**) box plots showing relative abundance of selected metabolites in LJ and LF groups. * *p* < 0.05, ** *p* < 0.01, *** *p* < 0.001, **** *p* < 0.0001

**Figure 6 foods-14-03684-f006:**
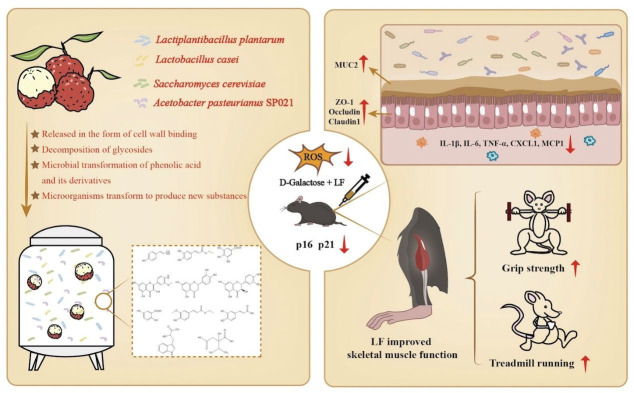
Schematic diagram of the mechanism of lychee fermentate on alleviating skeletal muscle and intestinal aging.

## Data Availability

The original contributions presented in this study are included in the article. Further inquiries can be directed to the corresponding authors.

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
