# Peer review of "Lychee Fermented by Mixed Probiotic Strains Alleviates D-Galactose-Induced Skeletal Muscle and Intestinal Aging in Mice"

_foods, 2025, doi:10.3390/foods14213684_

Round 1
Reviewer 1 Report
Comments and Suggestions for Authors
Han et al. investigated the protective effects of a fermented lychee product (LF) in a D-galactose-induced aging mouse model. This study aimed to evaluate the impact of LF on muscle function, intestinal integrity, oxidative stress, inflammation, and gut microbiota composition. The authors employed a comprehensive approach, including behavioral tests, histopathology, gene expression analysis, metabolomics, and 16S rRNA sequencing. This study addressed the relevant concept of the gut-muscle axis and positioned fermented plant products as a potential strategy to mitigate age-related physiological decline. This rationale is supported by the existing literature on fermentation-enhancing bioactive compound bioavailability and functionality. The scope of this work is impressive, spanning physiology, histology, molecular biology, metabolomics, and microbiology. The findings are novel and of potential interest to the readership of Foods. However, the manuscript should be revised as follows.
Major Comments
Lines 92-97, page 3: The justification for using the specific mixed-strain consortium (L. plantarum, L. casei, S. cerevisiae, A. pasteurianus) is somewhat weak. The authors state that mixed-strain fermentation is "more effective" but provide only a general citation (Ref 17). A more detailed rationale, perhaps hypothesizing on synergistic roles (e.g., yeasts breaking down complex sugars for lactobacilli and acetobacter potentially contributing to acid metabolism), would strengthen the study's conception.
Section 2.1: The initial microbial load (CFU/mL) of the inoculants was not specified, making the process difficult to reproduce.
Lines 111-112: The "0.05% (w/w)" inoculation for S. cerevisiae, L. plantarum, and L. casei is ambiguous. Is this 0.05% each or 0.05% total for the mixture? Clarification is required.
Lines 119-127: There was no group that received D-galactose + unfermented lychee juice (LJ). The LJ control was only used for metabolomics. Without this group, it is impossible to definitively attribute the observed effects to the fermentation process itself, rather than simply the lychee juice components. This is a major limitation of the study. The authors should explicitly acknowledge this limitation in the discussion. In future studies, an LJ + D-gal group will be critical. The discussion must temper claims about the unique benefits of fermentation versus lychee for this manuscript.
Lines 234-235, Figure 1B: The lack of effect on body weight is noted, but no data on food intake are provided. This is a common confounder in nutritional intervention trials. Either provide food intake data or acknowledge its absence as a limitation in the Discussion section.
Section 3.4 (Page 8-9): The gut microbiota analysis is comprehensive, but the description of the F/B ratio requires refinement. The interpretation of the Firmicutes/Bacteroidota (F/B) ratio is context-dependent and not a universal "health" indicator. The authors should be more cautious in their interpretation of the results. The authors may reframe the discussion of the F/B ratio, emphasizing the specific changes observed (increase in Firmicutes and decrease in Bacteroidota) and linking them to the specific beneficial genera that were enriched (e.g., Clostridia_UCG-014), rather than relying solely on the phylum-level ratio.
Section 3.5: Metabolomics data is a strength. However, the link between the metabolites identified in the LF and their in vivo effects is correlative. The discussion should more clearly state that while these metabolites are plausible candidates for bioactivity, their direct causal role in the observed physiological effects was not experimentally verified in this study.
Discussion: The discussion extensively covers the biotransformation of compounds but does not adequately address the major methodological limitation mentioned above: the absence of an unfermented lychee juice control group in the animal study. Please add a paragraph or significant text explicitly acknowledging this limitation. The conclusion that fermentation is responsible for the observed effects should be tempered, stating that the effects are attributed to "the lychee fermentate" and that future studies comparing it directly to the unfermented juice are needed to isolate the contribution of fermentation.
Page 13: The discussion on microbiota is well-linked to the results. The mention of KEGG pathways related to aging (Line 490-493) is interesting but underdeveloped. If possible, please provide a brief summary of the key aging-related pathways identified in Figure S4 (not provided) in the main text or supplementary results to provide more context for the reader.
Minor Comments
Line 123: “lowdose” should be correct to “low dose”.
Line 322: "D-galatpse" is a typo; correct to "D-galactose".
Scientific Nomenclature: Throughout the text, ensure consistency and correctness. For example, "Bifidobacterium animalis " (line 67) and other genus or species names should be italicized. Please check the entire manuscript.
Supplementary tables and figures mentioned in the text were not provided.
Author Response
Lines 92-97, page 3: The justification for using the specific mixed-strain consortium (L. plantarum, L. casei, S. cerevisiae, A. pasteurianus) is somewhat weak. The authors state that mixed-strain fermentation is "more effective" but provide only a general citation (Ref 17). A more detailed rationale, perhaps hypothesizing on synergistic roles (e.g., yeasts breaking down complex sugars for lactobacilli and acetobacter potentially contributing to acid metabolism), would strengthen the study's conception.
Response:
We thank the reviewer for this comment. In response, the rationale for using the specific mixed-strain consortium has been expanded in the revised manuscript (Line 97) as follows: “Furthermore, mixed-strain fermentation has been reported to be more effective than single-strain fermentation in promoting the release of phenolic compounds from lychee pulp and may yield enhanced biological activities [17-19]. During the fermentation process, yeasts catalyze the conversion of glucose and fructose into ethanol and carbon dioxide while simultaneously producing vitamins, amino acids, and other beneficial metabolites. Lactic acid bacteria utilize carbon sources and amino acids to generate a variety of bioactive substances, including organic acids, peptides, fatty acids, polysaccharides, and vitamins. Acetobacter strains oxidize alcohol to acetic acid and contribute to the production of antioxidant components such as polyphenols [20]”.
References:
- Tang, S., Luo, N., Zeng, Q., Dong, L., Zhang, R., He, S., Nag, A., Huang, F. and Su, D. Lychee pulp phenolics fermented by mixed lactic acid bacteria strains promote the metabolism of human gut microbiota fermentation in vitro. Food & Function 2023, 14, 7672-7681.
- Sheng, J., Shan, C., Liu, Y., Zhang, P., Li, J., Cai, W., Tang, F. Comparative evaluation of the quality of red globe grape juice fermented by Lactobacillus acidophilus and Lactobacillus plantarum. International Journal of Food Science and Technology 2022, 57, 2235-2248.
- Frey-Klett, P., Burlinson, P., Deveau, A., Barret, M., Tarkka, M., Sarniguet, A. Bacterial-fungal interactions: hyphens between agricultural, clinical, environmental, and food microbiologists. Microbiology and molecular biology reviews 2011, 75, 583-609.
- Yuan, X., Wang, T., Sun, L., Qiao, Z., Pan, H., Zhong, Y., Zhuang, Y. Recent advances of fermented fruits: A review on strains, fermentation strategies, and functional activities. Food Chemistry: X 2024, 22, 101482.
Section 2.1: The initial microbial load (CFU/mL) of the inoculants was not specified, making the process difficult to reproduce.
Response: We thank the reviewer for this comment. The manuscript has been revised to specify the initial microbial load of the inoculants. Section 2.1 now reads: "The juice was then inoculated with 0.05% (w/w) mixture containing S. cerevisiae, L. plantarum, and L. casei, to reach a final concentration of 5 × 10⁵ CFU/mL, 4 × 10⁵ CFU/mL, and 4 × 10⁵ CFU/mL, respectively." in the revised manuscript, Line 136-138.
Lines 111-112: The "0.05% (w/w)" inoculation for S. cerevisiae, plantarum and L. casei is ambiguous. Is this 0.05% each or 0.05% total for the mixture? Clarification is required.
Response: We have clarified this point in the revised manuscript. The sentence has been revised to: “The juice was then inoculated with 0.05% (w/w) mixture containing S. cerevisiae, L. plantarum and L. casei, to reach a final concentration of 5 × 10⁵ CFU/mL, 4 × 10⁵ CFU/mL, and 4 × 10⁵ CFU/mL, respectively.” (Lines 136-138)
Lines 119-127: There was no group that received D-galactose + unfermented lychee juice (LJ). The LJ control was only used for metabolomics. Without this group, it is impossible to definitively attribute the observed effects to the fermentation process itself, rather than simply the lychee juice components. This is a major limitation of the study. The authors should explicitly acknowledge this limitation in the discussion. In future studies, an LJ + D-gal group will be critical. The discussion must temper claims about the unique benefits of fermentation versus lychee for this manuscript.
Response: We thank the reviewer for highlighting this important point. n this study, our primary objective was to evaluate whether lychee fermentate (LF) could alleviate intestinal and skeletal muscle aging in a D-galactose-induced mouse model and to identify potential bioactive metabolites generated during multi-strain fermentation through untargeted metabolomics. Accordingly, different doses of LF were designed to assess its efficacy. However, we acknowledge that the animal experiment did not include a “D-galactose + unfermented lychee juice (LJ)” group, which limits our ability to distinguish the effects of fermentation-derived metabolites from those of the native lychee bioactives. Therefore, we have added a paragraph to the Conclusion section explicitly addressing this limitation and have tempered our claims in the Conclusion section. The limitation of this study has been included in the revised manuscript, Line 552-559.
Lines 234-235, Figure 1B: The lack of effect on body weight is noted, but no data on food intake are provided. This is a common confounder in nutritional intervention trials. Either provide food intake data or acknowledge its absence as a limitation in the Discussion section.
Response: The data on food intake have been added to the Supplementary Materials as Figure S1 (Average daily food intake of mice in each group during LF intervention). In the main text, the sentence has been revised to include this information: “Neither D-galactose injection nor LF supplementation significantly affected the body weight gain (Figure 1B) and daily food intake of mice in each group (Figure S1).” (Lines 259-261).
Section 3.4 (Page 8-9): The gut microbiota analysis is comprehensive, but the description of the F/B ratio requires refinement. The interpretation of the Firmicutes/Bacteroidota (F/B) ratio is context-dependent and not a universal "health" indicator. The authors should be more cautious in their interpretation of the results. The authors may reframe the discussion of the F/B ratio, emphasizing the specific changes observed (increase in Firmicutes and decrease in Bacteroidota) and linking them to the specific beneficial genera that were enriched (e.g., Clostridia_UCG-014), rather than relying solely on the phylum-level ratio.
Response: We thank the reviewer for this valuable comment and agree that the interpretation of the Firmicutes/Bacteroidota (F/B) ratio should be treated with caution, as it is context-dependent and not a universal indicator of gut health. Accordingly, we have revised the relevant sections to focus on the specific microbial changes observed rather than solely on the phylum-level ratio. Revisions have been made as follows:
HLF intervention reversed these alterations, increasing the relative abundance of Firmicutes and decreasing Bacteroidetes, bringing their proportions closer to the control group levels (Figure 4D). (Lines 359)
Lines 517 change: The relative abundance on phylum level in the HLF group was significantly increased in Firmicutes and decreased in Bacteroidetes compared to the D-gal group....
Lines 545 change: Furthermore, LF modulated gut microbiota composition, restoring the relative abundance of Firmicutes and Bacteroidetes, and enriching beneficial microbial taxa, which.....
Section 3.5: Metabolomics data is a strength. However, the link between the metabolites identified in the LF and their in vivo effects is correlative. The discussion should more clearly state that while these metabolites are plausible candidates for bioactivity, their direct causal role in the observed physiological effects was not experimentally verified in this study.
Response: We appreciate the reviewer’s insightful comment. We fully agree that the metabolomics data in this study establish a correlative rather than causal relationship between the metabolites identified in the lychee fermentate (LF) and the observed in vivo effects. Accordingly, we have revised the Discussion to explicitly acknowledge this limitation and to clarify that the identified metabolites are potential bioactive candidates whose direct roles require further experimental validation. The following statement has been added to the revised manuscript (Lines 552-559).
Discussion: The discussion extensively covers the biotransformation of compounds but does not adequately address the major methodological limitation mentioned above: the absence of an unfermented lychee juice control group in the animal study. Please add a paragraph or significant text explicitly acknowledging this limitation. The conclusion that fermentation is responsible for the observed effects should be tempered, stating that the effects are attributed to "the lychee fermentate" and that future studies comparing it directly to the unfermented juice are needed to isolate the contribution of fermentation.
Response: We thank the reviewer for highlighting this important point. n this study, our primary objective was to evaluate whether lychee fermentate (LF) could alleviate intestinal and skeletal muscle aging in a D-galactose-induced mouse model and to identify potential bioactive metabolites generated during multi-strain fermentation through untargeted metabolomics. Accordingly, different doses of LF were designed to assess its efficacy. However, we acknowledge that the animal experiment did not include a “D-galactose + unfermented lychee juice (LJ)” group, which limits our ability to distinguish the effects of fermentation-derived metabolites from those of the native lychee bioactives. Therefore, we have added a paragraph to the Conclusion section explicitly addressing this limitation and have tempered our claims in the Conclusion section. The limitation of this study has been included in the revised manuscript, Line 552-559.
Page 13: The discussion on microbiota is well-linked to the results. The mention of KEGG pathways related to aging (Line 490-493) is interesting but underdeveloped. If possible, please provide a brief summary of the key aging-related pathways identified in Figure S4 (not provided) in the main text or supplementary results to provide more context for the reader.
Response: We appreciate the reviewer’s valuable suggestion. To provide more context, a brief summary of the key aging-related KEGG pathways has been added to the revised manuscript (Lines 525-531), as “KEGG pathway enrichment analysis further indicated that D-galactose disrupted microbial metabolic pathways associated with energy and lipid metabolism while upregulating disease-related pathways. High-dose lychee fermentate (HLF) reversed these alterations, particularly downregulating aging- and inflammation-related pathways and restoring metabolic homeostasis, suggesting a protective modulation of gut microbiota function against aging-associated dysbiosis (Figure S4).”
Line 123: “lowdose” should be correct to “low dose”
Response: “lowdose” in the original manuscript was changed to “low dose” in the revised manuscript (Line 149).
Line 322: “D-galatpse” is a typo; correct to “D-galactose”.
Response: “D-galatpse” in the original manuscript was changed to “D-galactose” in the revised manuscript (Line 353).
- Scientific Nomenclature: Throughout the text, ensure consistency and correctness. For example, "Bifidobacterium animalis " (line 67) and other genus or species names should be italicized. Please check the entire manuscript.
Response: We appreciate the reviewer’s reminder regarding the consistency of scientific nomenclature. The genus and species names, including Bifidobacterium animalis, have been properly italicized in the revised manuscript (Line 70). A thorough check has also been performed to ensure that all scientific names throughout the text follow the correct formatting conventions.
Supplementary tables and figures mentioned in the text were not provided.
Response:We will submit the supplementary materials along with this submission.

Reviewer 2 Report
Comments and Suggestions for Authors
This paper investigates the beneficial effects of lychee fermentate (LF) produced by mixed probiotic strains on skeletal muscle and intestinal aging in a D-galactose mouse model. The study is timely, well-structured, and relevant to the field of functional foods, probiotics, and healthy aging. The experiments are comprehensive (muscle function, intestinal barrier integrity, inflammation, oxidative stress, microbiota, metabolomics). The findings are promising and support the role of LF as a functional food candidate. However, this work presents some imperfections according to the following comments:
Comments to Authors:
- The introduction provides a good background, but the novelty of this work compared to previous studies on lychee fermentation should be emphasized more clearly.
- The paper claims to offer “new insights,” but several cited studies have already reported enhanced antioxidant/anti-inflammatory effects of fermented lychee. Please clarify the unique contribution (mixed strains? gut–muscle axis focus? metabolomics integration?).
- Only male mice were used. Please justify why females were excluded, as sex differences may influence aging and microbiota.
- The study duration (8 weeks) is appropriate, but the dose selection rationale for LF (26.1 vs. 78.3 mg/kg) is not clearly explained.
- The fermentation protocol refers to a patent, but sufficient details should be provided in the manuscript for reproducibility.
- In the microbiota section, sequencing depth and rarefaction curves are not shown; these are essential for validating diversity analyses.
- The metabolomics results are rich, but the discussion should better integrate which specific metabolites are most relevant to the observed physiological effects.
- The English is generally understandable but needs polishing for grammar and flow (e.g., “D-gal injection and LF supplementation didn’t significantly affect…” → “Neither D-galactose injection nor LF supplementation significantly affected…”).
Author Response
The introduction provides a good background, but the novelty of this work compared to previous studies on lychee fermentation should be emphasized more clearly.
Response:
Thank you for this valuable suggestion. In the revised manuscript, we have further clarified the novelty of our study. Specifically, unlike previous reports focusing mainly on single-strain fermentation or general compositional changes in fermented lychee products, our work highlights the use of a multi-strain co-fermentation system (including Lactiplantibacillus plantarum, Lacticaseibacillus casei, Saccharomyces cerevisiae, and Acetobacter pasteurianus) to enhance the functional potential of lychee. Furthermore, we demonstrate for the first time that the resultant lychee fermentate alleviates intestinal and skeletal muscle aging in a D-galactose-induced murine model and employ untargeted metabolomics to elucidate the key bioactive constituents responsible for these effects. This section has been rewritten in the revised manuscript, Line 106-112.
The paper claims to offer “new insights”, but several cited studies have already reported enhanced antioxidant/anti-inflammatory effects of fermented lychee. Please clarify the unique contribution (mixed strains? gut–muscle axis focus? metabolomics integration?).
Response: Unlike prior studies that mainly assessed general antioxidant or anti-inflammatory outcomes, we specifically evaluated the impact of fermented lychee on skeletal muscle aging and its connection with intestinal health, providing mechanistic insights into the gut–muscle axis. Moreover, we combined biochemical, histological, microbiome and metabolomics analyses to characterize both systemic and local effects, allowing us to identify potential metabolite-mediated mechanisms underlying the observed protective effects. To facilitate better understanding for readers, this sentence has been rewritten as “In summary, these findings provided new insights into the potential application of LF as a functional food for mitigating skeletal muscle and intestinal aging” in the revised manuscript, Line 41-43.
Only male mice were used. Please justify why females were excluded, as sex differences may influence aging and microbiota.
Response: Only male mice were used in this study for the following reasons. First, female mice experience periodic fluctuations in sex hormones, which can confound results in aging and gut microbiota research (Zhang et al., 2018). Second, the D-galactose-induced aging model is more commonly applied in male mice, providing well-established, reproducible outcomes for systemic aging studies.
References:
- Zhang, H., Wang, Z., Li, Y., Han, J., Cui, C., Lu, C., Su, X. Sex-based differences in gut microbiota composition in response to tuna oil and algae oil supplementation in a D-galactose-induced aging mouse model. Frontiers in Aging Neuroscience 2018, 10, 187.
The study duration (8 weeks) is appropriate, but the dose selection rationale for LF (26.1 vs. 78.3 mg/kg) is not clearly explained.
Response: The dose selection of Lychee Fermentate (LF) in this study was primarily based on previously published research investigating the bioactive properties of lychee fermentation products. Specifically, we referred to the study by Huang et al. (2024), which demonstrated that lychee fermentate alleviated DSS-induced colitis by enhancing intestinal barrier function and modulating gut microbiota. The effective gavage dose used in their study served as a direct reference for determining the low- and high-dose regimens in our experimental design. In addition, minor adjustments were made based on our preliminary experimental experience to ensure suitability for the D-galactose-induced mouse model.
The dose selection of Lychee Fermentate (LF) has been included in the revised manuscript, Line 150.
References:
- Huang, R., Yao, H., Ji, S., Wu, J., Lin, Q., Gupta, T.B., Gan, D., Wu, X. A lychee fermentate with enriched acetate but lowered GABA attenuates DSS-induced colitis by reinforcing gut barrier function and modulating intestinal microbiota. Food Bioscience 2024, 59, 104089.
The fermentation protocol refers to a patent, but sufficient details should be provided in the manuscript for reproducibility.
Response: The details of fermentation has been included in the revised manuscript, Line 133, as “Fresh lychee fruits were washed thoroughly and pressed to obtain lychee juice. The fermentation process followed a patented two-stage protocol (ZL202210565739.1). Firstly, lychee juice (including pulp) was supplemented with sucrose to a final concentration of 0.15% (w/v). The juice was then inoculated with 0.05% (w/w) mixture containing S. cerevisiae, L. plantarum and L. casei, to reach a final concentration of 5 × 105 CFU/mL, 4 × 105 CFU/mL, and 4 × 105 CFU/mL, respectively. The first stage fermentation was conducted at 30°C for 72 h. Secondly, the fermented liquid was filtered, and the filtrate underwent secondary fermentation with 0.3% (w/v) A. pasteurianus SP021 at 30°C for an additional 72 h. Finally, the product was sterilized by pasteurization (80°C for 30 min) to obtain LF. Lychee juice (LJ) processed under identical pasteurization conditions served as a control”.
In the microbiota section, sequencing depth and rarefaction curves are not shown; these are essential for validating diversity analyses.
Response: We thank the reviewer for this valuable comment. In the revised manuscript, the sequencing depth and rarefaction curve analysis have been added to ensure the validity of the diversity analyses. Specifically, Table S2 (“Statistics of Sequence Information After Denoising”) and Figure S5 (“Rarefaction analysis of partial 16S rRNA gene sequences to estimate the microbial diversity of each sample”) have been included in the Supplementary Materials. In addition, the following sentence has been added to the revised manuscript, Line 346, as: "A total of 575,359 high-quality reads corresponding to 10,421 ASVs at single-nucleotide resolution were obtained, with an average of 31,964 reads per sample, which were suitable for downstream analysis of the gut microbial community (Table S2 and Figure S5)."
The metabolomics results are rich, but the discussion should better integrate which specific metabolites are most relevant to the observed physiological effects.
Response: We thank the reviewer for this insightful suggestion. In response, we have substantially revised the Discussion section to better integrate the metabolomics findings with the observed physiological outcomes. The revised discussion (Line 487) now highlights the specific metabolites most closely associated with the attenuation of intestinal and skeletal muscle aging. The revised section reads as follows:
"Flavonoids and phenolic acids generated during fermentation are widely recognized for their potent anti-inflammatory and antioxidant activities, which contribute to the attenuation of skeletal muscle and intestinal aging [27–29]. Among these, compounds such as isorhamnetin, eriodictyol, syringic acid, and quercetin have demonstrated significant efficacy in alleviating intestinal injury or muscle atrophy [30–36]. Notably, isorhamnetin enhances nuclear factor erythroid 2–related factor 2 (Nrf2) activity, upregulating antioxidant enzymes such as heme oxygenase-1 (HO-1), thereby protecting against intestinal inflammation [32]. Moreover, it inhibits TNF-α-induced apoptosis in HUVECs by suppressing NF-κB activation [37]. Similarly, gallic acid mitigates pentachlorophenol-induced intestinal damage through antioxidant mechanisms [38], while p-hydroxybenzoic acid has been reported to strengthen intestinal barrier function in mice [39].
In addition, lychee fermentate was enriched in several microbial-specific metabolites with anti-inflammatory properties. For instance, indole-3-lactic acid (ILA), which markedly increased after fermentation, is known to alleviate intestinal inflammation by downregulating pro-inflammatory cytokines such as TNF-α through activation of the aryl hydrocarbon receptor (AhR) and Nrf2 pathways. These pathways are essential for maintaining intestinal homeostasis and redox balance. Specifically, ILA upregulates AhR-target genes (e.g., CYP1A1) and Nrf2-targeted genes (e.g., GPX2 and SOD2), thereby conferring protection against oxidative stress and intestinal senescence [40,41]. Collectively, these metabolites suggest that the beneficial effects of fermented lychee are driven by the synergistic actions of flavonoids, phenolic acids, and microbial-derived compounds that jointly contribute to anti-inflammatory and antioxidant defense mechanisms."
This revision explicitly links the key metabolites identified by metabolomics to the physiological improvements observed, thereby strengthening the mechanistic interpretation of the findings.
The English is generally understandable but needs polishing for grammar and flow (e.g., “D-gal injection and LF supplementation didn’t significantly affect…” → “Neither D-galactose injection nor LF supplementation significantly affected…”).
Response: We appreciate the reviewer’s suggestion regarding language refinement. The sentence “D-gal injection and LF supplementation didn’t significantly affect” has been revised to “Neither D-galactose injection nor LF supplementation significantly affected” to improve grammatical accuracy and fluency (Line 259). In addition, the entire manuscript has been carefully polished for English grammar, clarity, and overall readability.
Reviewer 3 Report
Comments and Suggestions for Authors
In this interesting study, the authors employed a mixed culture fermentation strategy to
obtain Lychee fermentate. Furthermore, the research explored its potential in mitigating intestinal and muscle aging, supported by a detailed metabolomic analysis. The manuscript presents evidence-based scientific results, thereby demonstrating clear scientific value. The discussion section is also well-structured and thorough. However, to further enhance the quality of the present research, the following issues need to be addressed:
1) Page 2, Line 81. Please provide more detailed description explaining the previosly studied
bioactives of the most abudant bioactive compounds present in Lychee.
2) The subsection Reagents and Chemical is missing. Please include this subchapter along
with necessary details.
3) Page 3, Line 126. Please explain how was the Lychee fermentate administered orally i.e.
what solvent was used and what was the solubility of the Lychee fermentate? Please
include these details in the text.
4) On the basis of which the doses of Lychee fermentate within the low-dose and high-dose
regimen were selected? Based on previously published research or based on experimental
experience? Please include these details in the text.
5) Subchapter names in the results section are inadequate. Change the names of subsections
3.1-3.5. to sound a little more general. Namely it is not necessary to emphasize the
results/effects in the title of the subsection within Results.
6) At the end of the Conclusion, clearly state the main limitations of the current research.
7) The number of references is limited for a study of this scope. It is recommended to expand
the reference list by incorporating additional key and recent literature relevant to this field

Author Response
1) Page 2, Line 81. Please provide more detailed description explaining the previosly studied bioactives of the most abudant bioactive compounds present in Lychee.
Response: We thank the reviewer for this constructive suggestion. In the revised manuscript, we have expanded the background information to include a more detailed description of the previously studied bioactive compounds in Lychee chinensis. The text on Page 2, Line 84 has been revised as follows: "Lychee (Litchi chinensis) is rich in various bioactive compounds, particularly polyphenols such as proanthocyanidins, catechins, and gallic acid. Lychee has also been traditionally used as an analgesic in ancient Chinese medicine".
The subsection Reagents and Chemical is missing. Please include this subchapter along with necessary details.
Response: The section 2.1 “Materials and Main Reagents” has been included in the revised manuscript as “Lychee fruits (cv. Heiye) were purchased from the local market. Lactiplantibacillus plantarum and Lacticaseibacillus casei were purchased from Shanghai Jiaoda Onlly Co., Ltd., China. Saccharomyces cerevisiae used in this study was instant dry yeast, which was purchased from Yinglian Food Supplements Co., Ltd., China. Acetobacter pasteurianus SP021 was purchased from the China General Microbiological Culture Collection Center (CGMCC), with the collection number 25794. D-galactose was purchased from Sigma-Aldrich (St. Louis, MO, USA). Hematoxylin and Eosin (H&E) staining kit and Alcian Blue Periodic Acid Schiff (AB-PAS) staining kit were purchased from Solarbio (Beijing, China). Trizol reagent was purchased from Cwbio (Taizhou, China). Qubit RNA BR Assay Kit was acquired from Invitrogen (Eugene, Oregon, USA). HiScript III RT SuperMix and ChamQ SYBR Master Mix were purchased from Accurate Biology (Changsha, China). Primary antibody against MUC2 or ZO-1 was purchased from Proteintech (Wuhan, China). Primary antibody against Claudin-1 or Occludin, seondary antibody conjugated with Alexa Fluor 488 was purchased from Abcam (Shanghai, China). D-lactic acid assay kit and GSH-Px assay kit were purchased from Jiancheng Bioengineering Institute (Nanjin, China). MDA assay kit and SOD assay kit were purchased from Solarbio (Beijing, China) ”.
Page 3, Line 126. Please explain how was the Lychee fermentate administered orally i.e. what solvent was used and what was the solubility of the Lychee fermentate? Please include these details in the text.
Response: According to the preparation method of the lychee fermentate, the final product was obtained as a liquid solution. Therefore, it was directly administered to mice by oral gavage without the need for additional solvents or dilution.
On the basis of which the doses of Lychee fermentate within the low-dose and high-dose regimen were selected? Based on previously published research or based on experimental experience? Please include these details in the text.
Response: The dose selection of Lychee Fermentate (LF) in this study was primarily based on previously published research investigating the bioactive properties of lychee fermentation products. Specifically, we referred to the study by Huang et al. (2024), which demonstrated that lychee fermentate alleviated DSS-induced colitis by enhancing intestinal barrier function and modulating gut microbiota. The effective gavage dose used in their study served as a direct reference for determining the low- and high-dose regimens in our experimental design. In addition, minor adjustments were made based on our preliminary experimental experience to ensure suitability for the D-galactose-induced mouse model.
The dose selection of Lychee Fermentate (LF) has been included in the revised manuscript, Line 150.
References:
- Huang, R., Yao, H., Ji, S., Wu, J., Lin, Q., Gupta, T.B., Gan, D., Wu, X. A lychee fermentate with enriched acetate but lowered GABA attenuates DSS-induced colitis by reinforcing gut barrier function and modulating intestinal microbiota. Food Bioscience 2024, 59, 104089.
Subchapter names in the results section are inadequate. Change the names of subsections 3.1-3.5. to sound a little more general. Namely it is not necessary to emphasize the results/effects in the title of the subsection within Results.
Response: In accordance with the comment, we have revised the subchapter names to sound more general. For example, Section 3.1 has been changed to “The effects of LF on skeletal muscle function” and Section 3.4 to “The Modulatory Effects of LF on Gut Microbiota” in the revised manuscript.
At the end of the Conclusion, clearly state the main limitations of the current research.
Response: We thank the reviewer for this suggestion. In response, we have explicitly added a statement regarding the main limitation of the current study at the end of the Conclusion section. The added text reads as follows: "The primary limitation of this study is the absence of a “D-galactose + unfermented lychee juice” control group in the animal experiments, as unfermented lychee juice was included only for metabolomic analyses. This design prevents definitive attribution of the observed benefits—such as enhanced antioxidant activity and modulation of gut microbiota—to fermentation-derived metabolites versus the intrinsic bioactive compounds present in lychee juice, including phenolics and polysaccharides. Consequently, future studies directly comparing fermented and unfermented lychee with this mouse model are warranted."
The number of references is limited for a study of this scope. It is recommended to expand the reference list by incorporating additional key and recent literature relevant to this field
Response: We appreciate the reviewer’s valuable suggestion regarding the limited number of references. We fully agree that, given the scope and depth of this study, additional references are necessary to strengthen the academic foundation of the work. Accordingly, we have updated the reference list in the revised manuscript by incorporating several recent and highly relevant studies that align closely with our research focus. These additions enhance the scientific rigor and contextual depth of the paper. For example, the following references have been added:
- Sheng, J., Shan, C., Liu, Y., Zhang, P., Li, J., Cai, W., Tang, F. Comparative evaluation of the quality of red globe grape juice fermented by Lactobacillus acidophilus and Lactobacillus plantarum. International Journal of Food Science and Technology 2022, 57, 2235–2248.
19. Frey-Klett, P., Burlinson, P., Deveau, A., Barret, M., Tarkka, M., Sarniguet, A. Bacterial–fungal interactions: hyphens between agricultural, clinical, environmental, and food microbiologists. Microbiology and Molecular Biology Reviews 2011, 75, 583–609.
20. Yuan, X., Wang, T., Sun, L., Qiao, Z., Pan, H., Zhong, Y., Zhuang, Y. Recent advances of fermented fruits: A review on strains, fermentation strategies, and functional activities. Food Chemistry: X 2024, 22, 101482.
Round 2
Reviewer 1 Report
Comments and Suggestions for Authors
The authors have done an excellent job revising the manuscript. The revised version is significantly improved, and I recommend it for acceptance.
Reviewer 2 Report
Comments and Suggestions for Authors
The authors answered all questions. They also improved their manuscript. I recommend publishing this manuscript in this form.